# Real-Time System for Driver Fatigue Detection Based on a Recurrent Neuronal Network

**DOI:** 10.3390/jimaging6030008

**Published:** 2020-03-04

**Authors:** Younes Ed-Doughmi, Najlae Idrissi, Youssef Hbali

**Affiliations:** 1Department Computer Science, FST, University Sultan Moulay Sliman, 23000 Beni Mellal, Morocco; n.idrissi@usms.ma; 2Computer Systems Engineering Laboratory Cadi Ayyad University, Faculty of Sciences Semlalia, 40000 Marrakech, Morocco; youssef.hbali@gmail.com

**Keywords:** driver fatigue detection, drowsiness, recurrent neural networks

## Abstract

In recent years, the rise of car accident fatalities has grown significantly around the world. Hence, road security has become a global concern and a challenging problem that needs to be solved. The deaths caused by road accidents are still increasing and currently viewed as a significant general medical issue. The most recent developments have made in advancing knowledge and scientific capacities of vehicles, enabling them to see and examine street situations to counteract mishaps and secure travelers. Therefore, the analysis of driver’s behaviors on the road has become one of the leading research subjects in recent years, particularly drowsiness, as it grants the most elevated factor of mishaps and is the primary source of death on roads. This paper presents a way to analyze and anticipate driver drowsiness by applying a Recurrent Neural Network over a sequence frame driver’s face. We used a dataset to shape and approve our model and implemented repetitive neural network architecture multi-layer model-based 3D Convolutional Networks to detect driver drowsiness. After a training session, we obtained a promising accuracy that approaches a 92% acceptance rate, which made it possible to develop a real-time driver monitoring system to reduce road accidents.

## 1. Introduction

The World Health Organization (WHO) has identified road traffic injuries as a major global public health problem. As indicated in the World Road Safety Status Report 2015 [1], they are one of the major causes of death and injury. Each year, nearly 1.25 million persons die, and millions more are injured or handicapped as a result of road traffic accidents. Figure 1 shows the upward trend in the number of road deaths between 2001 and 2013.

A study conducted in Morocco [2] in 2013 demonstrated a worrying number of accidents identified with drowsiness, which is responsible every year for more than 4000 deaths and 1.4 billion dollars of material harm. Early drowsiness accounted for 36.8% of cases, and drowsiness was accounted for in 31.1% of cases, including the quarter of the month preceding the survey. Likewise, 42.4% of drivers did not meet the suggested break each 500 km or 2 h of driving in Figure 2.

Given the earnestness of the issue, the academic community and concerned industry accept that has become essential to concentrate on setting up a compelling framework to avoid the beginning of exhaustion and diminish the number of mishaps, wounds, and deaths brought about by this phenomenon.

Based on our previous work [3] in the same area of drowsiness detection with deep learning, in the current paper, we use another model of Recurrent Neural Networks (RNN) that we will detail in the following sections—in particular that these models are well-performing and powerful at the calculation level [4,5,6], which allows us to save time in the training process because this step is often costly in time and money.

For this, we used the dataset Drowsy Driver Detection (NTHU-DDD) [7] used in the work [8]. The NTHU-DDD video data set consists of five different scenarios. The data set is comprised of a group of male and female drivers from different ethnic groups. Each frame in the video labeled with “fatigue” or “not fatigue.” Videos consist of different types of “fatigue” and “not fatigue” activities, including day and night. Videos are 640 × 480 pixels, 30 frames per second in Audio Video Interleave (AVI) format without audio. Our technique plans to improve the precision of drowsiness identification through binary classification, and the dataset NTHU-DDD contains black and white pictures of a few classes identified with drowsiness. This enables us to perform supervised learning computations on the drowsiness part to accomplish optimal learning conditions for a higher achievement rate. Next, the dataset NTHU-DDD was split into small batches of 7 s each that contain the necessary action that best reflects drowsiness.

However, research has focused on the physical, psychological, and physiological state of the driver, such as the study on the influence of the driver’s motivation and emotion on his decision-making ability [9]. In addition, the environment surrounding the driver is a factor that influences the driver’s degree of concentration and acts on his nervousness [10]. In addition, many methods related to the study of driver fatigue have been proposed in the literature [11,12,13,14]. However, these methods analyze the driver’s state via sensors installed either on the car or the driver directly. These techniques give very optimal results. On the other hand, our data source is different because we perform image processing to extract a driver’s behavior. The combination of these two solutions can give higher results.

To recap, the objective of this work is to improve road wellbeing conditions and anticipate deadly mishaps. In this way, we propose a variation of the RNN calculation that has just demonstrated its effectiveness on drowsiness detection, which is distinguishing an activity-dependence on a sequence. This work allows the car business to utilize this kind of system to give reliable, simple, and reasonable arrangements.

## 2. Related Work

The objective of the different techniques used in the field of driver drowsiness detection is to represent and detect the signs of the driver’s drowsiness. This technique is based on the behavior and appearance of the driver or both.

Many studies [15,16,17,18] have been conducted on the state of the driver’s eye based on a video surveillance system to detect the eyes and calculate their frequency of blinking to check the driver’s fatigue level. Some studies [18], such as Adaboost’s [18], have used a cascade of classifiers for the rapid detection of the ocular area. Thus, in our work, we use an RNN model [15], which aims to analyze the slow and long closure of the driver’s eyes. This approach uses a Haar-like descriptor [19] and an AdaBoost classification algorithm [20] for face and eye-tracking by using Percent Eye closure (PERCLOS) to evaluate driver tiredness. PERCLOS evaluates the proportion of the total time that a drivers’ eyelids are ≥80% closed and reflects a slow closing of the eyelids rather than a standard blink.

There is also research on separate head-pose estimation of the driver’s head [21]. The Separate Head Position Estimation System is a method that results in the orientation of the head relative to a camera view. The objective is therefore detected if the driver’s head is inclined forward, which is indicative of a loss of concentration. To this, a specific descriptor is created in this research by merging four descriptors of the highly pertinent head orientation, and one finds the steerable filters, the histogram of oriented gradients, Haar features [19], and an adapted version of the Speeded-Up Robust-Features descriptor. Second, these features apply in the classification algorithm of the support vector machine [22].

Other studies were conducted on the health of drivers [14,23,24,25,26], in order to analyze when the driver is no longer able to drive and detect the first signs of fatigue for anticipating the onset of drowsiness to alert the driver that it is necessary to take a break. For this type of method, it uses Electroencephalography (EEG), which makes measurements using the captures pasted directly on the skin of the driver by measuring heart rate, body temperature, and other measurements depending on the type of work. In addition, another type of capture is installed in a car directly; for example, it captures the measurements of the temperature inside and outside the car. All of these calculated measurements are used to feed a vector model, which classifies the measured values between alert and extreme fatigue states; this type of work gives good results. Still, industrial-scale use, especially for ordinary drivers, is costly and not practical for everyday use.

Work is also carried to detect drowsiness [27,28,29]. The solution proposed is a combined system for fatigue detection and the detection of unwanted driver behavior during driving. Using the Kinect for WindowsDeveloper Toolkit, Kinect has made it possible to detect simple postures by detecting the position of the driver’s body limbs and analyzing their movement to detect the postures that are closest to a driver’s fatigue behavior.

Finally, in-depth learning is a trend that is increasing across general data analysis and identifies as one of the top 10 technologies of 2018 [30]. Several works are using deep learning [31,32], such as deep learning being an upgrade of artificial neural networks. It consists of combining a large number of layers that allow for a higher level of abstraction and improved predictions from the often complex and heterogeneous data.

## 3. Background

### 3.1. Deep Learning

Deep learning is an upgrade of artificial neural networks, made up of a more layered structure allowing higher levels of abstraction and better predictions from the data [33,34]. Deep learning is always developing into the primary machine learning tool in the general imaging and computer vision domains.

The concept of deep learning provides several algorithms and methods that give multitudes of possibilities to extract certain characteristics of drowsiness, For extracting the intermediate representation, algorithms consist of proprietary software [35,36,37], or a pre-trained CNN [4] such as the VGG-16 one [5]. For characterizing drowsiness, models consist of logistic regression [36], artificial neural network (ANN) [35,38], support vector machine (SVM) [39], hidden Markov model (HMM) [8], Multi-Timescale by CNN [15], long-short term memory (LSTM) network smoothed by a temporal CNN [4], or end-to-end 3D-CNN [6] are conducted.

### 3.2. Recurrent Neural Networks (RRN)

Late advances in profound learning have made it conceivable to remove significant level highlights from crude information, prompting leaps forward in computer vision [40,41,42].

RNN is innately somewhere down in time as their concealed state is an element of all past hidden states, as shown in Figure 3.

The simplest recursive network comprises a repeating unit connecting an input vector $\left(x_{1},x_{2}\ldots ,x_{T}\right)$, a hidden layer vector of vectors $\left(h_{1},h_{2},\ldots ,h_{T}\right)$, and an output vector $\left(y_{1},y_{2},\ldots ,y_{T}\right)$. The representations are created by the nonlinear transformation of the input sequence from t= 1 to *T*. Some simple recurrent networks (SRNs) include the Elman network [43] and Jordan network [44].

However, the RNN used in other research problems, such as [45], was utilized for the generative adversarial networks to generate abstract text summarization. NNH-CNN [46] applied local convolutional filters with max-pooling to the frequency instead. We also found P-CNN [47] and Auto-Conditioned LSTM [48], which are methods for recognizing and predicting actions. There is also Ref. [49,50] on predicting the 3D shape and texture of objects from a single image. In [51], the authors proposed a dilated convolutional neural network to capture time dependencies in the context of driver maneuver prediction. All of these methods show the multitude of choices for dealing with a sequential problem. Our choice was motivated by the result published in the previous work of the 3D ConvNets method, which shows a very high predictive efficiency.

In the following section, we present the multi-layer architecture based on an RNN model, which may be an optimal solution to the problem discussed in this work.

## 4. Proposed Approach

This section presents the proposed multi-layer network architecture. The basic model was described first and then a detailed description of the layers were chosen for our multi-layer architecture.

### 4.1. Learning Features with 3D Convolutional Networks

The proposed 3D Convolutional Networks (3D ConvNets) method [52] learns 3D ConvNets (Figure 4) on a limited temporal support of 16 consecutive frames with filter kernels of size 3×3×3, and the choice of kernel size has the objective of reducing the weight generated by the model to use it on equipment with limited resources. [53] It was shown that dynamic neural networks used to address time-dependent behavior run significantly faster than similar techniques such as the Hidden Markov model. The authors have demonstrated that HMM computational resources are impractical for mobile and embedded systems. Better performance than that in [54] was reported by allowing all filters to operate over space and time. However, the network is considerably deeper than [54,55], with a structure similar to the profound networks such as [5]. Another way of learning spatiotemporal relationships was proposed in [56], in which the authors factorized a 3D convolution into a 2D spatial convolution and a 1D temporal convolution. Specifically, their temporal convolution factorized into a 2D convolution over time as well as feature channels, and it is used only at higher layers of the network.

### 4.2. Multi-Layer Architectures

The global multi-layer architecture of the suggested drowsiness detection based on the ConvNets 3D model [52]. From this model, we have tried to establish the best profile because it considered one of the most accurate models for similar problems. The next paragraphs explain the architecture and the layers used in deep learning.

According to the results realized by the 3D ConvNets, the best option for homogeneous adjustment with convolution cores is 3×3×3; these cores are as deep as possible. An example can be derived using large-scale data.

Furthermore, note that the 3D ConvNet architecture contains several deep layers that explain the duration of computing, which is rather long but converges quickly from the first epoch. Thus, in the multi-layer architecture, there are six layers for the 3D model ConvNets, four layers max-pooling [57,58], one flatten layer [59], and two fully connected layers, followed by a Softmax [60,61] output layer (Figure 5).

The architecture multi-layers allow us to reduce the complexity of the data each time we move from one layer to another layer. Such 3D ConvNets layers perform a filter that passes over the image sequence, scanning a small number of pixels at a time and generating a feature map that predicts the class to which each feature relates. At the entrance of the first layer, we start with a dimension of 32×32 pixels. After allowing it to process the selected zone of the image quickly, and to predict its class, each passage to another layer 3D ConvNets the dimension scanned by the model multiplied by 2, which gives 256×256 pixels for the last layers 3d ConvNets; this augmentation of the filtered zone augments the precision of the prediction. Then, at each pooling layer, it reduces the size of the information for each characteristic obtained in the 3D ConvNets layer while retaining essential information. After filter step 3D ConvNets and the data reduction with the max-pooling layer, the flatten layer takes the output of the previous layers, flattens them, and transforms them into a single vector, which could be an input for the next step. We have used max-pooling as a reduction method because of the nature of the images that have a gray background. The fully connected layers continue to use input from the characteristic analysis and apply weights to predict the right label. Finally, we find the layer of the Softmax activation function, which gives us the final class of the enter sequence. In our case, it is a binary classification.

## 5. Experiments

### 5.1. The NTHU-DDD Dataset

We used the NTHU-DDD dataset of drowsiness detection to learn about driver drowsiness with a different subject (see Figure 6). This dataset implements five scenarios for each class study—for example, how one finds driving with and without standard glasses in the day and at night. In addition, we analyzed the case of driving with sunglasses in the day (see Figure 7), which is a different scenario—the figures show all possible cases for drowsiness of a driver.

Thus, we split all videos according to wanted activity in order to obtain the sequences of the drowsiness class only. We selected two clips from every video, and every one of the clips is a maximum of 7 s long. Then, we provided another sub-set of data that represents the regular comportment of the driver. Finally, we obtained 849 clips. This filtering of videos from the dataset chosen in preference to best extract sequences that indicate drowsiness and to exclude sequences that may prejudice learning scores.

### 5.2. Training

The preparation of the NTHU-DDD dataset starts by cutting recordings of 36 subjects, as depicted in the preceding subsection, to make two learning classes with the end goal that the first-class relates to drivers’ ordinary conduct, and the second class relates to the drivers’ drowsiness. All recordings were isolated into three classifications: 60% for training, 30% for validation, and 10% for testing.

After that, based on the selected architecture, the frame sizes of each clip were adjusted to optimize the size of the machine’s memory.

To execute this training, we used a PC with Alienware R17, Ubuntu 16.04 LTC, 16 GB RAM, and 8 GB GPU. The software part consists of the Keras framework [62] with the Python programming language, such that Keras is a library providing predefined methods for deploying deep learning models using a low-level training backend; for our case, we use TensorFlow [63].

After training, a test step was performed on a dataset not used during the training phase in order to validate the accuracy of the produced model. Table 1 shows the results of the training performed on several calculation models compatible with the training of the video sequences.

To illustrate how 3D ConvNets works internally, we used the de-convolution method [64] using the useful tf_cnnvis [65] that extracts images from the deconvolution. Figure 8 visualizes the deconvolution of a drivers figure case in drowsiness, in which the function focuses on the whole person, and then follows the leaning of the person’s head in the remainder of the images.

## 6. Results Analysis

In this section, we present the results of the training of our model under many measures and presentation. Thus, with an acceptance rate converging to 97% and an error rate converging to 5.00% over 100 epochs, we can observe that an architecture based on the 3D ConvNets model gives good results compared to sequential solutions. For this, we used other models to enrich the result tables. We find LSTMs [66], Long-term recurrent convolutional networks (LRCNs) [67], Hierarchical Temporal Deep Belief Network (HTDBN) [8], Deep Belief Nets (DBN) [68] + SVN [39], and Multilayer Perceptron Neural Network (MLP) [69].

### 6.1. Accuracy Results

Table 1 illustrates the rate of the calculations performed on the different computing models with the accuracy and validation rates, and the test rate performed on unused clips in the training phase.

In addition, testing in different sequences shows that it is necessary to test with different numbers of images per sequence for each problem and to choose the best sequence according to the efficiency and speed of the model. In our case, 40 images per sequence give a better result for our problem with the conv3d 3D model.

### 6.2. Evolution of the Training

Figure 9 illustrates the convergence of the training result of the 3D ConvNets architecture. As shown, the acceptance rate increases rapidly in the first iteration and stabilizes at the end of the learning. However, as this architecture is memory intensive, it utilizes more time to finish the learning phase, and the learning time varies according to the number of layers chosen for the 3D ConvNets architecture and the size of the dataset.

### 6.3. Performance Measurement

In order to measure the performance of our training, a test phase was carried out with data not used in the training and validation phase, and the results obtained in the test phase are very encouraging.

Table 2 presents a $F_{1}$ score [70] analysis of the binary classification, the $F_{1}$ score is a measure of precision of a test, and it takes into account both the precision p and the recall r of a test to calculate the score. In the table, we present the different F1 score measures realized for the D ConvNets model, which shows an encouraging precision that exceeds 80.

### 6.4. Comparison with Other Methods

Compared with [8], in which somnolence experiments conducted on a dataset. Our proposed model shows an optimal result on the same dataset for the classification of somnolence.

Table 3 shows the effectiveness of applying different basic models (SVM, DBN, HMM, or HTDBN) and new technologies that use deep neural networks.

## 7. Proposed System

In this section, we propose a software architecture for a mobile application illustrated in Figure 10, in which the mobile application will use the phone’s camera to capture a sequence of frames at a frequency of five frames a second. Through the drowsiness prediction model, computations will be performed in real-time; if the model predicts drowsiness, a visual and audio message will run on the user’s phone.

In a second step, and with the agreement the user, for each detected prediction, a feedback message will be requested to the user to label the sequences of images used. Then, a transfer will be executed to a web server, which retrieves the labeled images. These images will be saved and analyzed for the prediction module to trained. The objective of this step is to ensure that the system’s maintenance is scalable and to improve the accuracy of the model’s prediction.

The technical stack proposed to implement our software architecture is described below:Mobile application: Kivy and Tensorflow (Python3)Web server: Flask and Tensorflow (Python3).Database: Postgresql (sql).Storage: File system or cloud.

## 8. Conclusions

In this study, we have shown how using deep learning multi-layer architectures RNN using 3D ConvNets, trained on a large scale video dataset, can help solve sequential problems such as the one discussed in this paper, that is to say, the detection and prevention of drowsiness.

Our goal in this work was to provide a neural network architecture for an affordable and portable sleepiness detection service for ordinary drivers. The ConvNet 3D architecture achieved accuracy close to 97% on all data used. To further improve these results, a more customized data set needs to be created that is more appropriate to the subject of drowsiness in an environment close to what the driver may experience in a real-life scenario.

On the other hand, our study was based on the driver’s behavior, which took place over some time with a change in the driver’s posture. This action is limited in the case where the driver is drowsy without changing his posture. Our solution cannot predict this particular case, but it can improve with an association with studies conducted on the driver’s physiology [74].

## Figures and Tables

**Figure 1 jimaging-06-00008-f001:**
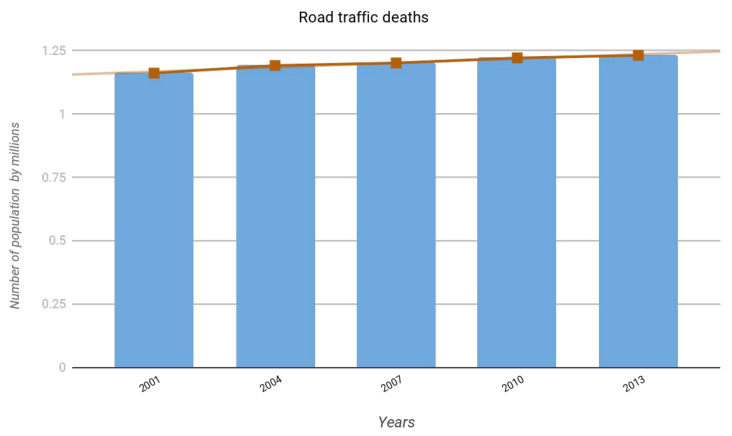
Road deaths on the world’s—World Health Organization (WHO) report 2015 [1].

**Figure 2 jimaging-06-00008-f002:**
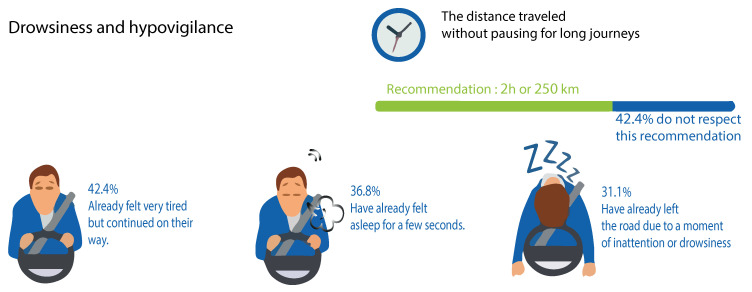
Study of prevalence and risk factors for drowsy driving in a Moroccan population.

**Figure 3 jimaging-06-00008-f003:**
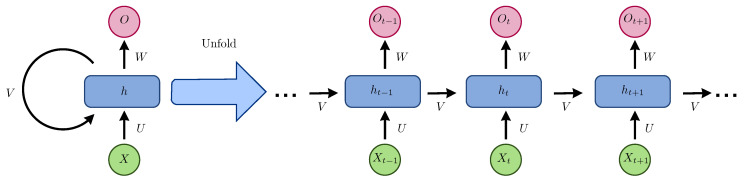
Recurrent neural network and the unfolding in the computation time involved in its forward computation.

**Figure 4 jimaging-06-00008-f004:**
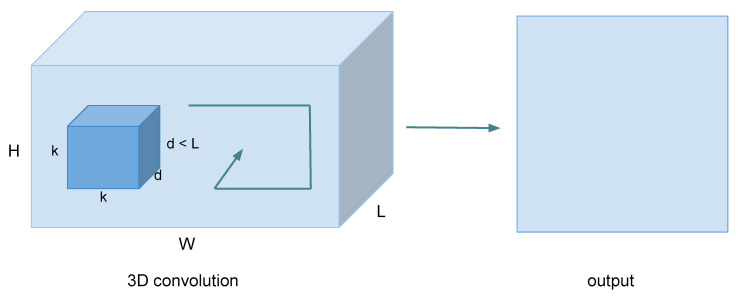
3D convolution operations; the three directions (x,y,z) represent the calculation of a convolution. The output shape represents a 3D Volume; input = [W,H,L] and filter = [k,k,d]; then, output = [W,H,M], and most importantly, d<L for obtaining output volume.

**Figure 5 jimaging-06-00008-f005:**
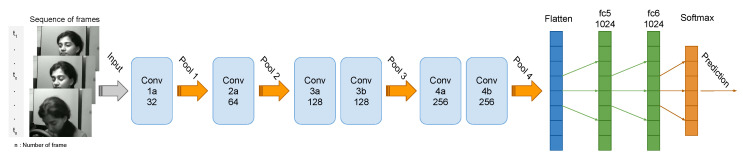
Proposed multi-layer architectures.

**Figure 6 jimaging-06-00008-f006:**
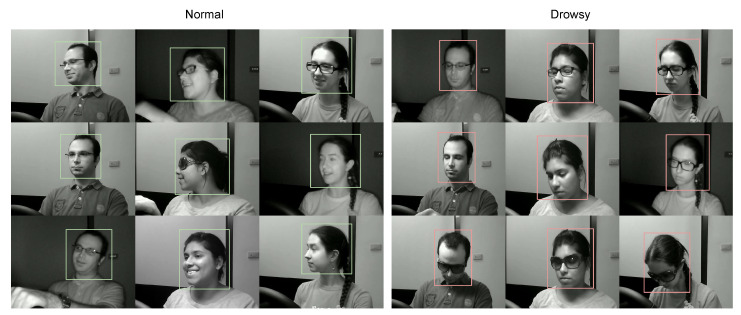
Samples of the two classes, normal and drowsy, of the NTHU-DDD dataset.

**Figure 7 jimaging-06-00008-f007:**
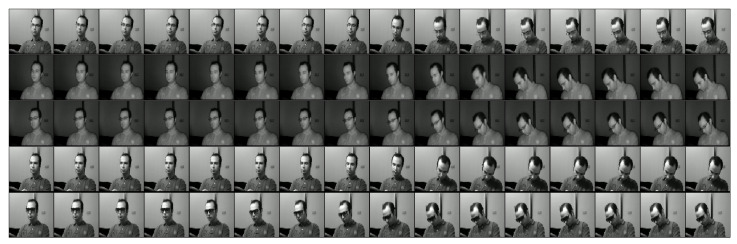
Various sequences of a subject with different postures from the NTHU-DDD dataset.

**Figure 8 jimaging-06-00008-f008:**
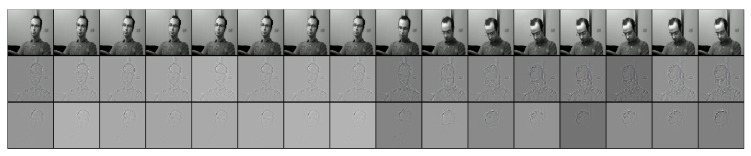
Application of the learned filters of the conv4b layer of our model; using the method from [64] with library tf_cnnvis [65] for extracting deconvolution images.

**Figure 9 jimaging-06-00008-f009:**
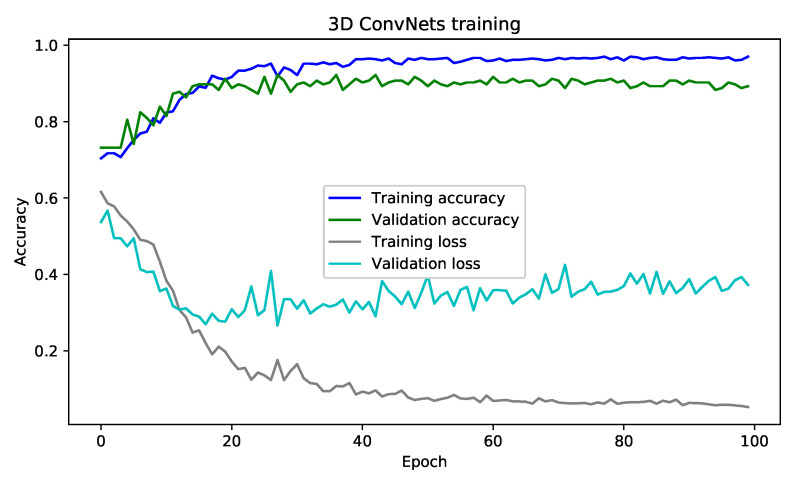
Result training for 3D ConvNets.

**Figure 10 jimaging-06-00008-f010:**
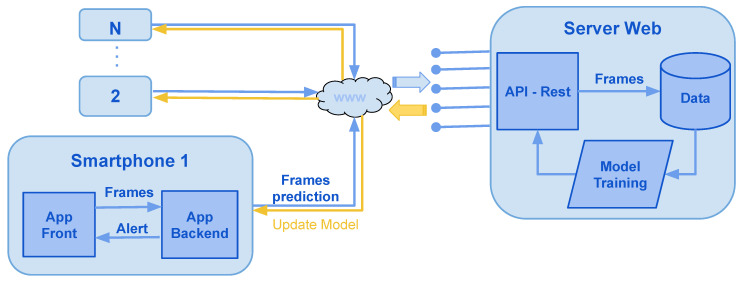
Software architecture for a mobile application and web server.

**Table 1 jimaging-06-00008-t001:** Drowsy driving detection on the NTHUDDD dataset.

Model	Sequence	Accuracy %	Validate %	Test %
Proposed method	20	97.00	92.19	78.05
	30	97.30	90.19	73.17
	40	97.12	90.40	82.00
LSTMs [66]	20	92.51	90.07	80.36
	30	92.58	90.06	78.04
	40	92.71	90.01	80.36
LRCN [67]	20	91.18	82.44	78.04
	30	90.72	81.86	78.04
	40	90.84	80.80	78.04
HTDBN [8]	40	83.04	82.65	80.44
	30	83.26	81.25	78.47
	40	83.04	82.65	80.44
DBN+SVM [39,68]	20	80.65	80.41	76.51
	30	80.01	78.58	75.21
	40	81.12	80.75	76.73
MLP [69]	20	71.71	73.17	60.97
	30	71.33	73.04	60.97
	40	71.18	72.22	60.97

**Table 2 jimaging-06-00008-t002:** $F_{1}$ Score experimental results.

Model	Sequence	Precision %	Recall %	*F*_1_ Score %
Proposed method	20	74	100	85
	30	72	92	81
	40	72	92	81
LSTMs [66]	20	100	62	77
	30	100	44	61
	40	100	62	77
LRCN [67]	20	75	96	80
	30	80	96	82
	40	75	96	80
HTDBN [8]	20	71	94	79
	30	68	94	77
	40	71	94	79
DBN+SVM [39,68]	20	60	84	64
	30	58	84	64
	40	60	84	61
MLP [69]	20	61	100	76
	30	61	100	76
	40	61	100	76

**Table 3 jimaging-06-00008-t003:** Comparison of 3D ConvNets and baseline solution on scenarios in the drowsy-driver-detection dataset.

Method	Accuracy %	*F*_1_ Scrore %
Driver Alertness Monitoring [71]	77.40	43.3
Embedded Smart Cameras [72]	81.40	43.7
HTDBN [8]	84.82	79.0
Proposed method	**92.19**	85.0
Multi-timescale CNN [15]	94.22	-
HDMS [73]	96.10	81.8

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
