# Peer review of "Real-Time System for Driver Fatigue Detection Based on a Recurrent Neuronal Network"

_2313-433X, 2020, doi:10.3390/jimaging6030008_

Round 1

Reviewer 1 Report

Good:

The paper discusses an important aspect, namely driver distraction and its resultant fatalities.

The focus on using RNN is a good approach to provide a potential for a realistic solution to detecting distraction.

To be improved:

The language should be more moderate (by removing strong bias words like significant, disturbing...) and replacing them with more moderate claims.

The numbers given for fatalities and cost are in Morocco and in local currency. I suggest to expand the review to other countries and use a conversion to euros or dollars (or at least give a conversion rate) to help international readers understand better.

Author Response

We would like to thank you for your efforts and your comments which have helped us to improve our work.

Point 1: The language should be more moderate (by removing strong bias words like significant, disturbing...) and replacing them with more moderate claims.

Response 1:Thank you for your remark, we have updated the article to improve the sentences and connecting words.

Point 2: The numbers given for fatalities and cost are in Morocco and in local currency. I suggest to expand the review to other countries and use a conversion to euros or dollars (or at least give a conversion rate) to help international readers understand better.

Response 1:We would like to thank the reviewer for this comment. We have added to the first paragraph of the introduction section the road deaths statistics worldwide And we have to update the currency exchange rate.

Reviewer 2 Report

The paper provides just a limited comparison against results of prior methods. The English usage is awkward at various places throughout the paper, and the grammar needs some correction. Also, the capitalization is incorrect in the paper title and in the column headings of Table 3, as well as at the beginning of par. 3 of the “Conclusions” section. The English usage is sometimes vague, as in the following examples: “After the planning of the dataset, the learning was started utilizing the calculation explained in the section below.“ “There is also work being done to detect [23–25].” How can the authors determine generalizability to other datasets if their network was tested on only one dataset? “Datasets that corresponds well to the driver’s real context are difficult to find or have difficult access rights, the dataset we used [6], is intended to show that the proposed calculation model can provide good results and generalize on any other dataset.” Not sure what this means, what is the difference between condition and behaviour? “A large number of methods related to the study of driver fatigue have been proposed in the literature [8–10]. We followed this research to select a subset of sleepiness detector characteristics, but our objective is different because we perform image processing to analyze driver behaviour, while these methods analyze driver condition, which adds an additional complexity compared to our approach if you want to manufacture this type of a solution.“ If the intention of these techniques is to detect drowsiness, then the output of their method should indicate drowsiness. It is unclear whether the following statement refers to the authors’ proposed techniques. “The objective of the different techniques carried out is to detect the signs of drowsiness, these techniques are based on the behavior of the driver, on the appearance of the driver or both techniques together, these techniques often give us a result at a time t that is often exact, but the limit of these techniques is that it cannot tell us if the detection performed corresponds to a case of drowsiness or a normal case of the driver’s behavior, ...” The “Related Works” section could be more thorough. A reference citation must be given for the following statement. “The global multi-layer architecture of the suggested drowsiness detection is based on the convNets 3D model. From this model, we have tried to establish the best profile, because it is considered as one of the most accurate models for similar problems.” Just because the filter size 3x3x3 is shown to be optimum for the dataset used by ConvNet does not mean it will be the optimum filter size for this application. It seems like the network simply detects the orientation of the head (Figure 6) and, based on that, decides if the driver is feeling drowsiness. In order to better validate the network, the author should have added some samples of a fully conscious driver looking anywhere but straight to see if the network is able to distinguish between drowsiness and other states such as checking the phone or dashboard screen. In Table 2 it should be clarified whether these architectures were taken from existing methods. If so, then they require a reference citation to those works. Furthermore, if these architectures are not taken from existing methods, then the comment regarding comparison to other existing methods has not been addressed. This work seems to detect head movement rather than tracking eyelid state (Figure 6), which seems to be the main focus of other works. Perhaps if this is in fact the better approach, the authors should explain how they determined that head movement has a direct relation to eyelid closure state.

Author Response

We would like to thank you for your efforts and your comments which have helped us to improve our work.

Point 1: The paper provides just a limited comparison against results of prior methods.

Response 1: Thank you for your comment. I have updated the paper with more comparisons of recent related work.

Point 2: The English usage is awkward at various places throughout the paper, and the grammar needs some correction.

Response 2: Thank you for your comment. We've revised the grammar of the article, and I hope I've improved this part.

Point 3: The capitalization is incorrect in the paper title and in the column headings of Table 3, as well as at the beginning of par. 3 of the “Conclusions” section.

Response 3: We would like to thank the reviewer. we updated the article and we have made all the changes to correct all capitalization.

Point 4: The English usage is sometimes vague, as in the following examples: “After the planning of the dataset, the learning was started utilizing the calculation explained in the section below.“

Response 4: We have reformulated this paragraph to better explain the idea.

Point 5: “There is also work being done to detect [23–25].” How can the authors determine generalizability to other datasets if their network was tested on only one dataset?

Response 5: Thank you for your pertinent remark. Unfortunately the only public dataset that we found available for drowsiness detection is the NTHU-DDD, we hope that in the future more public datasets will be available.

Point 6: “Datasets that corresponds well to the driver’s real context are difficult to find or have difficult access rights, the dataset we used [6], is intended to show that the proposed calculation model can provide good results and generalize on any other dataset.” Not sure what this means, what is the difference between condition and behaviour?

Response 6: Thank you for your pertinent remark. Unfortunately the only public dataset that we found available for drowsiness detection is the NTHU-DDD, we hope that in the future more public datasets will be available.

Point 7: “A large number of methods related to the study of driver fatigue have been proposed in the literature [8–10]. We followed this research to select a subset of sleepiness detector characteristics, but our objective is different because we perform image processing to analyze driver behaviour, while these methods analyze driver condition, which adds an additional complexity compared to our approach if you want to manufacture this type of a solution.“ If the intention of these techniques is to detect drowsiness, then the output of their method should indicate drowsiness. It is unclear whether the following statement refers to the authors’ proposed techniques.

Response 7: The idea behind this paragraph was to show that methods using different data than images to classify the driver's state give optimal results. However, the sensors needed to capture these data should installed on the car or the driver itself. I have reformulated the paragraph to clarify this point.

Point 8: “The objective of the different techniques carried out is to detect the signs of drowsiness, these techniques are based on the behavior of the driver, on the appearance of the driver or both techniques together, these techniques often give us a result at a time t that is often exact, but the limit of these techniques is that it cannot tell us if the detection performed corresponds to a case of drowsiness or a normal case of the driver’s behavior, ...” The “Related Works” section could be more thorough. A reference citation must be given for the following statement.

Response 8: We have rewritten this paragraph to cite articles related to this field.

Point 9: “The global multi-layer architecture of the suggested drowsiness detection is based on the convNets 3D model. From this model, we have tried to establish the best profile, because it is considered as one of the most accurate models for similar problems.” Just because the filter size 3x3x3 is shown to be optimum for the dataset used by ConvNet does not mean it will be the optimum filter size for this application.

Response 9: Of course, you are right, but the choice of the size 3*3*3 kernel made to reduce the weight generated by the model for using in equipment with limited resources.

Point 10: It seems like the network simply detects the orientation of the head (Figure 6) and, based on that, decides if the driver is feeling drowsiness. In order to better validate the network, the author should have added some samples of a fully conscious driver looking anywhere but straight to see if the network is able to distinguish between drowsiness and other states such as checking the phone or dashboard screen.

Response 10: We agree with you. For this reason, I added figure 6, which reflects the driver's posture.

Point 11: In Table 2 it should be clarified whether these architectures were taken from existing methods. If so, then they require a reference citation to those works. Furthermore, if these architectures are not taken from existing methods, then the comment regarding comparison to other existing methods has not been addressed.

Response 11: Thank you for your comment, the architectures taken from existing methods. We have added a reference citation for each method.

Point 12: This work seems to detect head movement rather than tracking eyelid state (Figure 6), which seems to be the main focus of other works. Perhaps if this is in fact the better approach, the authors should explain how they determined that head movement has a direct relation to eyelid closure state.

Response 12: Effectively, in the drowsy class, we selected the video sequences that show a movement of the head with partial or total closure of the eyes, to increase the probability of detecting the first signs of drowsiness.

Reviewer 3 Report

This paper presents a machine learning method for driver fatigue detection using neural networks. The paper addresses an interesting detection issue, and it looks like such methods will have applicability to the more intelligent cars we will be using in the near future. The authors describe with a lot of details their initial thoughts and the reasons why this study is important. In addition, the paper includes results that might be useful for those interesting in fatigue recognition. At this point, I would like to make some recommendations to the authors in order to improve their paper and make it ready for a journal publication.

Comment #1

I think the abstract exceeds the 200-word standard. Therefore it should be reduced.

Comment #2

The lack of theoretical rationale and the need for such detection methods are significant omissions in the introduction section. Although the introduction well addressed the motivation of the study, the paper fails to promote itself. I was thinking that you might need to consider discussing a bit about prior studies that relate to the need of understanding why we need to know the physical, psychological or even physiological state of car driver. These few paragraphs can be motivated using the below papers:

--Towards understanding motivational and emotional factors in driver behaviour: Comfort through satisficing

--The Effects of Driving Habits on Virtual Reality Car Passenger Anxiety

--Measuring neurophysiological signals in aircraft pilots and car drivers for the assessment of mental workload, fatigue and drowsiness

Comment #3

I would like to comment that there are other techniques for dimensionality reduction such as PCA, LLE, etc. Did you experiment with them? What is the advantage of your method if no dimensionality reduction technique is used? Moreover, instead of reducing the dimensionality of the input did you experiment to preprocess the input data. It is known that Restricted Boltzmann Machines (RBMs) can be used to preprocess the data and basically to help the "machine learning" process become more efficient. I would suggest the authors discuss in the paper about such a possibility as well as to consider a future implementation where RBMs will be used. Additional evaluation between the current method and RBM would make the paper even stronger. Papers that should be added and discussed:

-- A global geometric framework for nonlinear dimensionality reduction

-- Evaluating the covariance matrix constraints for data-driven statistical human motion reconstruction

-- A comparison of PCA, KPCA and ICA for dimensionality reduction in support vector machine

-- Laplacian eigenmaps for dimensionality reduction and data representation.

-- Multimodal deep learning

-- Learning Sparse Feature Representations for Music Annotation and Retrieval

-- Learning Motion Features for Example-Based Finger Motion Estimation for Virtual Characters

Comment #4

In this paper, the method evaluated using images. What are the potentials of such a project? Can we use such a method using other forms of data captured during the driving activity e.g., the hand pressure on the steering wheel? the authors should discuss the actual potentials of such a method by giving examples that such a method can be applied. This would also provide readers with the ability to understand the potentials of such a method.

Comment #5

In this paper, the authors implemented NNs. However, there is a number of machine learning techniques out there that might benefit in such an implementation. Thus, I would like to ask the authors to discuss the potential implementation of the proposed method to other machine learning techniques. Examples include hidden Markov models (HMM), Convolutional NN, Dilated Convolutional NN, regression, and graphical models. I would suggest the authors discuss the possible usage of the proposed DNN as well as about the advantages and disadvantages when implemented DNN to other methods. Here I suggest a few papers that use different machine learning techniques to solve computational problems 

-- Generative Adversarial Network with Policy Gradient for Text Summarization

-- Applying convolutional neural networks concepts to hybrid NN-HMM model for speech recognition

-- P-cnn: Pose-based cnn features for action recognition

-- Auto-Conditioned LSTM Network for Extended Complex Human Motion Synthesis

-- Photorealistic facial texture inference using deep neural networks. 

-- Dilated Convolutional Neural Network for Predicting Driver's Activity

-- Deep convolutional neural networks for detecting secondary structures in protein density maps from cryo-electron microscopy

Given all raised issues, I think a major revision is required. I believe that after the authors addressing the raised comments, the paper will be ready for publication.

Author Response

We would like to thank you for your efforts and your comments which have helped us to improve our work.

Point 1: I think the abstract exceeds the 200-word standard. Therefore it should be reduced.

Response 1: Effectively the abstract was a little long, and I reduced the abstract to respect the standards of many words while preserving the global understanding of the abstract.

Point 2: The lack of theoretical rationale and the need for such detection methods are significant omissions in the introduction section. Although the introduction well addressed the motivation of the study, the paper fails to promote itself. I was thinking that you might need to consider discussing a bit about prior studies that relate to the need of understanding why we need to know the physical, psychological or even physiological state of car driver. These few paragraphs can be motivated using the below papers:

  • Towards understanding motivational and emotional factors in driver behaviour: Comfort through satisficing.
  • The Effects of Driving Habits on Virtual Reality Car Passenger Anxiety.
  • Measuring neurophysiological signals in aircraft pilots and car drivers for the assessment of mental workload, fatigue and drowsiness.

Response 2: Thank you for your suggestion, which I found very relevant and which adds another aspect of the literary element to my work, for which I have added a discussion on this type of research in the introduction paragraph 7.

Point 3: I would like to comment that there are other techniques for dimensionality reduction such as PCA, LLE, etc. Did you experiment with them? What is the advantage of your method if no dimensionality reduction technique is used? Moreover, instead of reducing the dimensionality of the input did you experiment to preprocess the input data. It is known that Restricted Boltzmann Machines (RBMs) can be used to preprocess the data and basically to help the "machine learning" process become more efficient. I would suggest the authors discuss in the paper about such a possibility as well as to consider a future implementation where RBMs will be used. Additional evaluation between the current method and RBM would make the paper even stronger. Papers that should be added and discussed:

  • A global geometric framework for nonlinear dimensionality reduction
  • Evaluating the covariance matrix constraints for data-driven statistical human motion reconstruction
  • A comparison of PCA, KPCA and ICA for dimensionality reduction in support vector machine -- Laplacian eigenmaps for dimensionality reduction and data representation.
  • Multimodal deep learning
  • Learning Sparse Feature Representations for Music Annotation and Retrieval
  • Learning Motion Features for Example-Based Finger Motion Estimation for Virtual Characters

Response 3: Thank you for your remark. We have used max-pooling as a reduction method because of the nature of the images having a gray background, which results in more brightness for the subject, we have added an explanation of this choice in the "Proposed approach" section. For deep multimodal learning, we consider using a Kinect sensor in the future to training a 4D deep learning model. This solution will need an installation of the sensor on the car to collect first training data and then apply it to detect the driver's behavior.

Point 4: In this paper, the method evaluated using images. What are the potentials of such a project? Can we use such a method using other forms of data captured during the driving activity e.g., the hand pressure on the steering wheel? the authors should discuss the actual potentials of such a method by giving examples that such a method can be applied. This would also provide readers with the ability to understand the potentials of such a method.

Response 4: Using other sensors will surely make it possible to import data to detect drowsiness. The installation of these sensors in a car will only be a matter of cost. Therefore, to give an idea of an implementation of our system, we have added the section "System proposal" which illustrates a proposal of the system architecture and the description of the software environment.

Point 5: In this paper, the authors implemented NNs. However, there is a number of machine learning techniques out there that might benefit in such an implementation. Thus, I would like to ask the authors to discuss the potential implementation of the proposed method to other machine learning techniques. Examples include hidden Markov models (HMM), Convolutional NN, Dilated Convolutional NN, regression, and graphical models. I would suggest the authors discuss the possible usage of the proposed DNN as well as about the advantages and disadvantages when implemented DNN to other methods. Here I suggest a few papers that use different machine learning techniques to solve computational problems :

  • Generative Adversarial Network with Policy Gradient for Text Summarization
  • Applying convolutional neural networks concepts to hybrid NN-HMM model for speech recognition 
  • P-cnn: Pose-based cnn features for action recognition
  • Auto-Conditioned LSTM Network for Extended Complex Human Motion Synthesis
  • Photorealistic facial texture inference using deep neural networks.
  • Dilated Convolutional Neural Network for Predicting Driver's Activity
  • Deep convolutional neural networks for detecting secondary structures in protein density maps from cryo-electron microscopy

Response 5: Thank you for this analysis. We have improved the literature in the RNN subsection of the background section as well as we have enriched the list of methods that use RNN. On the other hand, we have discussed the choice of neural networks rather than the Hidden Markov model and other techniques in the "proposed approach" section.

Round 2

Reviewer 3 Report

After carefully reading the revised version of the paper as well as the responses made by the authors of this paper, I feel confident that this is a strong and scientifically sound paper. For this reason, I would like to recommend this paper for the Journal of Imaging. Well done!

This manuscript is a resubmission of an earlier submission. The following is a list of the peer review reports and author responses from that submission.